# Focus on Achalasia in the Omics Era

**DOI:** 10.3390/ijms251810148

**Published:** 2024-09-21

**Authors:** Anna Laura Pia Di Brina, Orazio Palmieri, Anna Lucia Cannarozzi, Francesca Tavano, Maria Guerra, Fabrizio Bossa, Marco Gentile, Antonio Merla, Giuseppe Biscaglia, Antonello Cuttitta, Francesco Perri, Anna Latiano

**Affiliations:** 1Division of Gastroenterology and Endoscopy, Fondazione IRCCS Casa Sollievo della Sofferenza, 71013 San Giovanni Rotondo, Italy; a.dibrina@operapadrepio.it (A.L.P.D.B.); o.palmieri@operapadrepio.it (O.P.); a.cannarozzi@operapadrepio.it (A.L.C.); f.tavano@operapadrepio.it (F.T.); mariaguerra.mg247@gmail.com (M.G.); f.bossa@operapadrepio.it (F.B.); m.gentile@operapadrepio.it (M.G.); antoniomerla@hotmail.it (A.M.); giuseppe.biscaglia@gmail.com (G.B.); f.perri@operapadrepio.it (F.P.); 2Unit of Thoracic Surgery, Fondazione IRCCS Casa Sollievo della Sofferenza, 71013 San Giovanni Rotondo, Italy; antonellocuttitta@gmail.com

**Keywords:** achalasia, omics, genomics, transcriptomics, microbiomics, viromics, proteomics

## Abstract

Achalasia is a rare and complex esophageal disease of unknown etiology characterized by difficulty in swallowing due to the lack of opening of the lower esophageal sphincter and the absence of esophageal peristalsis. Recent advancements in technology for analyzing DNA, RNA and biomolecules in high-throughput techniques are offering new opportunities to better understand the etiology and the pathogenetic mechanisms underlying achalasia. Through this narrative review of the scientific literature, we aim to provide a comprehensive assessment of the state-of-the-art knowledge on omics of achalasia, with particular attention to those considered relevant to the pathogenesis of the disease. The notion and importance of the multi-omics approach, its limitations and future directions are also introduced, and it is highlighted how the integration of single omics data will lead to new insights into the development of achalasia and offer clinical tools which will allow early diagnosis and better patient management.

## 1. Introduction

Achalasia, first described by Willis in 1674 as “food blockage in the esophagus”, is a rare motility disorder characterized by dysphagia, regurgitation, chest pain and weight loss, with the absence of coordinated contractions in the esophagus and the failure of the lower esophageal sphincter (LES) to relax due to the loss of inhibitory nitrinergic neurons in the esophageal myenteric plexus [1]. Based on a recent systematic review and meta-analysis, the global pooled incidence and prevalence of achalasia were estimated to be 0.78 cases per 100,000 person-years and 10.82 cases per 100,000 person-years, respectively, without significant difference according to sex, and higher incidence and prevalence in adults than in children [2]. Achalasia is categorized into three subtypes based on manometric patterns, clinical presentation and responses to treatments: type I with minimal contractility in the esophageal body and no pressurization, type II with intermittent periods of pan-esophageal pressurization and type III with premature or spastic distal esophageal contractions [3]. The most common form of achalasia is idiopathic (without a known or demonstrable cause) and mostly occurs sporadically, whereas in the minority of cases, it is a familial disorder that follows a dominant inheritance pattern [4], with the probability that there is a rare subform of early-onset recessively inherited [5]. For this reason, genetics could be a valid tool to identify in advance at-risk patients who could benefit from targeted therapies and to prevent or slow the progression of the disease [6]. That is why it may be essential to consider the family history of achalasia when evaluating an individual with the disease.

Achalasia is a heterogeneous condition with an etiology not fully understood. Autoimmune and inflammatory processes, possibly triggered by infectious agents, exposure to environmental factors, occurrence of familial achalasia, its association with well-defined genetic syndromes and a plethora of secondary causes, have been linked to the manifestation of the disease, and although there is a belief that genetic factors, along with their impacts on transcriptomics, microbiomics, viromics and proteomics may play a significant role, there is still a lack of concrete evidence [7].

Since the study of ”omics” is now widespread in the fields of biology and medicine and is no longer considered mere exploratory efforts, in this review we used idiopathic achalasia as an illustrative example of a complex clinical condition to delve into various aspects of omes, omics and multi-omics. Thus, omics technologies could represent a promising tool, in the future, for a comprehensive knowledge of the achalasia etiology and pathology.

For a more complete and understandable overview of omics on achalasia, we have added to each section the description of preliminary studies, through which, by applying less complex investigation strategies and genetic studies, have identified alleles, polymorphisms and common loci, genes, miRNAs, bacteria, viruses and proteins laying the foundations for future discoveries.

## 2. Omes, Omics and Multi-Omics

Basically all diseases, and not just so-called “complex diseases”, are multifaceted because countless interconnected biological components contribute to their pathophysiology [8]. Any pathology has an intricate mix of genetic, epigenetic, environmental and behavioral factors that determine their onset and progression. Consequently, it is evident that studying each component of the disease in isolation, although it gives valuable information on any markers or sheds light on the different biological pathways between patients with a disease and a control group, is not sufficient to understand the intricate biological mechanisms at play [9]. For example, analysis of a single gene can reveal mutations associated with a disease, but cannot explain the entire clinical picture without considering the interaction of that gene with other genes, proteins and metabolites within the biological network.

So the need to study all components in a much more comprehensive way has emerged, hence the emphasis on omes and omics, defined, respectively, as the object of study of such field and its analysis. Substantially, omics is intended to improve our understanding of human biology and promote health and well-being through biomedical science and research [10]. Examples of omics disciplines include the identification of genes (genomics), messenger RNA (transcriptomics), epigenetic factors (epigenomics), proteins (proteomics) and finally, metabolites (metabolomics).

Thus ideally, different technologies would be combined to help diagnose disease and to generate effective clinically actionable tools to aid medical decision-making, generating a holistic picture of human phenotypes and disease. Furthermore, the integration of various types of omics data, multi-omics and clinical data, enables the production of more detailed information and is commonly used to discover potential underlying changes that trigger diseases contributing to the understanding of the pathogenic mechanisms or to identify potential targets for treatments. Indeed, it is expected that omics data will help the development of new and effective drugs, aimed at identifying biomarkers, novel molecular or protein targets for disease-modifying therapies useful for early diagnosis, disease staging and prediction progression. The National Cancer Institute defined biomarkers as biological molecules in blood, bodily fluids, or tissues that reveal whether a process, condition, or disease is normal or aberrant. Proteins, nucleic acids, antibodies and peptides are only a few of the many molecules that make up a biomarker, but also gene expression patterns, proteomic profiles, and metabolomic signatures are a combinations of modifications they can include.

However, despite great progress in this field, precise and definitive omics-based disease biomarkers have yet to be found for the majority of disorders, including inflammatory and neoplastic conditions [9]. Investigations using technologies such as proteomics and DNA microarrays have yielded over 150,000 articles documenting thousands of claimed biomarkers, but fewer than 100 have been validated for routine clinical practice [11]. Examples of mutation- and gene alteration-based well-established cancer biomarkers are the kirsten rat sarcoma virus (*KRAS*) and B raf proto oncogene (*BRAF*) mutations, both confirmed prognosis biomarkers for anti-epidermal growth factor receptor (EGFR) monoclonal antibody therapy response for metastatic colorectal cancer [12], and breast cancer 1/2 (*BRCA1/BRCA2*) mutations which increase cancer risk in breast/ovarian cancer, and guide therapy selection [13]. MammaPrint is a gene-expression-based assay used to analyze the activity of a set of genes in breast cancer providing a genomic risk score that helps determine the risk of distant metastasis and assists in treatment decision-making, particularly in early-stage breast cancer [14]. In the comprehensive review paper by Das et al. [15], a variety of biomarkers applicable to cancer detection and diagnosis are described. Ideally, biomarker tests intended for use in patient care would undergo rigorous evaluation prior to introduction into the clinic, and it is important to consider the functional characterization through pathways and network analysis of specific biomarkers, along with its implementation feasibility in terms of public health. Successfully translating omics-based predictors into clinical practice requires a rigorous development and validation process that follows a checklist of criteria covering issues relating to specimen collection, processing and storage conditions, assay procedures, mathematical modeling, interpretation of the test result and ethical, legal and regulatory aspects [16]. All these steps raise specific challenges regarding validation practices and determine the use of these omics-based tests. However, with the goal of better defining what is intended for “biomarker”, and capturing distinctions between biomarkers and clinical assessments, the Food and Drug Administration and the National Institutes of Health working group identified many types of biomarkers with distinct roles in biomedical research, clinical practice and medical product development providing clarity in the biomarker field [17].

For the sake of completeness it is important to highlight that, notwithstanding the implementation of multi-omics data introduces new computational and interpretive opportunities, there are hidden pitfalls in the omics sciences, particularly those involving high-throughput omics data processing and interpretation, data privacy and security, legal and ethical considerations and the need to integrate omics data with other clinical information to gain insight more complete health of the patient. For example, big data generated by high-throughput omics technologies requires advanced computational infrastructures and sophisticated algorithms to manage, analyze and interpret, and sometimes the results are overinterpreted and not supported by high-quality data and analysis [18]. Protecting patient privacy is critical, as omics data can reveal sensitive information about genetic predispositions and health status [19]. Omics data collection and analysis require informed consent from participants, who must be adequately informed about the risks and benefits. Furthermore, the handling and storage of biological samples raise questions about how to ensure confidentiality. Omics data are highly sensitive, and carry the risk of re-identification, even when anonymized, threatening the privacy of patients; they are vulnerable to cyber-attacks, requiring robust security measures to protect them from unauthorized access. Other ethical and legal issues related to the use of omics data may include problems related to inequalities in access to personalized medicine due to its high cost; control over the data due to its unclear ownership; complications related to inconsistent international regulations, and ethical dilemmas in deciding whether and how to return omics data results to patients [20]. Importantly, the results of omics analyses can have psychological impacts on patients and their families, particularly if genetic predispositions to disease are revealed. Ensuring that clinicians can accurately interpret and apply these results in the context of individual patient care is crucial. The sensitive information should be communicated in a clear and careful way. Patients must be actively involved in the decision-making process, receiving information about the risks and benefits of the technologies used. This approach not only strengthens their trust in the system but also ensures ethical and respectful care for their needs. Finally, it is essential that legal policies protect patient rights, and promote the ethical use of data in this emerging field but, despite the progress in the field, there are still vulnerabilities in genetic databases, which makes continuous monitoring of data management practices necessary [21]. In addition, a number of other factors influence and modify the results, reproducibility and interpretation of multi-omics studies, highlighting the importance of considering them as critical variables in biological and clinical research, as they can reveal crucial information for understanding and prevention. One example is the environment to which the patient was exposed at birth, and later throughout his or her following life. Other examples of modifying factors include the patients’ diet, their stress status and also their sex. Biological differences between males and females, including variations in genetic characteristics, hormone levels and transcriptome levels, can lead to differences in gene expression profiles, protein production and genetic regulation thus influencing patients’ metabolism and immune response [22].

Hence, although omics sciences offer enormous potential for personalized medicine and disease prevention, it is essential to address and overcome these challenges to effectively translate the results of omics research into concrete benefits for patients, by moving from a targeted and reductionist approach to a more global, and integrative one.

In this context, scientific evidence about the study of achalasia with an omics approach is scarce and only future studies will be able to elucidate new perspectives and open new avenues for the management of complex diseases like this. On a purely exploratory basis, we have looked into the achalasia from different omics perspectives to provide a good narrative review focusing on genomics (DNA), transcriptomics (RNA—microRNA), metagenomics (microbial DNA), viromics (viruses) and proteomics (proteins) [23] (Figure 1).

## 3. Search Methodology

We conducted a comprehensive literature search of the PubMed/MEDLINE database from inception to 30 May 2024, to identify studies reporting data on achalasia. The database was searched using the term achalasia, which was combined using the set operator “AND” with studies identified using the following terms: genetic, gene, polymorphism, DNA, HLA, susceptibility, pathogenesis, association, risk factor, RNA, regulation, miRNA, microbiota, bacteria, virus, infection, proteins, inflammation, cytokines and chemokines. All terms were used as medical subject headings terms. Restriction to the English language was applied. Two investigators (Anna Latiano and Anna Laura Pia Di Brina) independently assessed the eligibility of the articles for the study and extracted the data. After reviewing the titles and abstracts of all selected studies and the bibliographies of all the eligible papers, we examined the studies and divided the literature into five topics: genomics, transcriptomics, microbiomics, viromics and proteomics based on the specific focus. We excluded reviews and studies with non-human subjects (animal models). About the selected studies, there were no limitations on the number of subjects, methodology applied, analysis performed, treatment options and biological material used. Furthermore, no new data were created or analyzed in this review (Figure 2).

## 4. Genomics

In the area of medical research, genomics focuses on the structure and function of an organism’s whole DNA. Today, genomics with advancements in technology, DNA sequencing and computational analysis plays a significant role in helping to understand, prevent and provide opportunities to personalize patient treatment approaches.

Among the omics, genomics is the most mature on idiopathic achalasia, though it has not established conclusive links to specific genes so far (Table 1).

Several studies report that achalasia is significantly associated with specific alleles of the human leukocyte antigen (*HLA*) class II system. The first study, using serological identification techniques, showed that individuals with DQw1 antigen had 4.2 times more risk of developing the disease than the rest of the population [24]. Subsequently, using molecular biology techniques, other results showed an association between the achalasia and the allele HLA-DQA1*0101 and two HLA-DQαβ heterodimers [25], DQB1*0602 [26], DQA1*0103 and DQB1*0603 [27], HLA-DP and HLA-DR [28]. With the advent of new technologies for finding disease-susceptibility genes, a research team conducted the first systematic genome-wide association study (GWAS), a research approach that explores hundreds of thousands of genetic variants in the entire genome of large numbers of individuals to identify genomic variants that are statistically associated with a risk for a disease. Genotyping was performed on the Immunochip, covering 196,524 polymorphisms (SNPs) at immune-related loci throughout the genome, in 1068 individuals with achalasia and 4242 controls of central European ancestry, identifying a marked linkage signal in the major histocompatibility complex region on chromosome 6 by imputing classical HLA haplotypes and amino acid SNPs. Among the 33 markers identified, an eight-residue insertion at position 227-234 (rs28688207) in the cytoplasmic tail of HLA-DQβ1 emerged as the variant most closely related to achalasia, conferring an increased risk of developing the disease (*p* = 1.73 × 10^−19^) [[29], Figures 1 and 2]. In an extensive epidemiological investigation and genotype-phenotype analysis, Becker et al. revealed that patients carriers of the eight-residue HLA-DQβ1 insertion had a higher incidence of viral infections before achalasia onset, allergies and autoimmune disorders, and that their first-degree relatives showed a similar prevalence of these conditions. In particular, it has emerged that hormonal and immunosuppressive factors can be a trigger for achalasia in pregnant women carrying the polymorphism [30]. Likewise, it was demonstrated that the insertion in HLA-DQβ1 also conferred achalasia risk in the Polish and Swedish populations, with a geospatial north-south gradient in Europe [[31], Figure 1]. Next, a cross-sectional retrospective, genotype-phenotype association study conducted by Vackova et al. investigated the frequency of rs28688207 insert and revealed heterogeneity in achalasia patients concerning the high-resolution manometry subtype, with a more significant association in subtype I, suggesting a specific immune-mediated role for this subtype in the pathogenesis of the disease [32]. Another study by Janette Furuzawa-Carballeda et al. described, using high-resolution HLA typing, the distribution of HLA alleles and the conserved extended haplotypes in a group of Mexican mixed-ancestry patients finding a significant increase in the alleles frequencies of DRB1*14:54 and DQB1*05:03 and the extended haplotypes, and of DRB1*11:01-DQB1*03:01 in patients with achalasia compared to controls, revealing that the HLA-DRB1*14:54-DQB1*05:03 haplotype was introduced by admixture with European and/or Asian populations [33]. De la Concha et al. explored the presence of the HLA class II alleles and tumor necrosis factor-α (TNFα) and TNFβ microsatellites in patients with achalasia, observing that the TNFα11 allele and the DRB1*1501-DQA1*0102-DQB1*0602 haplotype were less common in patients, strengthening the possible relationship between the disease and the HLA-DQ1 allele and indicating that TNFα11 could be associated with protection from the pathology [34]. Additionally, in another study 391 SNPs encompassing 190 immune and 67 neuronal genes were genotyped in a large cohort of achalasia patients and healthy volunteers, identifying the SNP rs1799724, located on chromosome 6p21 between the lymphotoxin-α (*LTA*) and *TNFα* genes, as a susceptibility factor for idiopathic achalasia [[35], Table 1 and Figure 2].

Case-control association studies in achalasia analyzed polymorphisms in genes involved in the regulation of immune responses. The first one aimed to investigate the role of Arg381Gln (rs11209026) SNP, located on the interleukin-23 receptor (*IL23R)* gene, in the Spanish population, showing an increase in minor allele frequency in patients compared with healthy controls, specifically in male patients with disease onset after 40 years [36]. Based on the previous data Nuñez et al. genotyped the *IL10* rs1800896, rs1800871 and rs1800872 promoter SNPs reporting that the GCC haplotype influenced the risk of achalasia in the Spanish population [37]. Subsequently, a panel of eleven SNPs in the *IL33*, *IL1RL1*, *IL23R* and *IL10* genes was analyzed in an Italian cohort, using TaqMan genotyping assays, reporting significant differences in allele and genotype frequencies of the rs3939286 variant of the *IL33* gene between achalasia patients and controls [38]. On the other hand, Santiago’s group performed a case-control study genotyping, by TaqMan chemistry, the functional C1858T (rs2476601) polymorphism in the protein tyrosine phosphatase N22 (*PTPN22*) gene encodes a lymphoid-specific phosphatase, a downregulation of T-cell activation, observed significant differences between the analyzed cohorts with an increased risk of developing achalasia in Spanish women. Moreover, not significantly different in pre- and post-menopausal women were pointed out, emphasizing that hormonal levels did not seem to be responsible for gender bias [39].

**Table 1 ijms-25-10148-t001:** Comprehensive overview of genomic studies on achalasia.

Study	Gene	Main Finding	Methods
Wong RK, et al., 1989 [24]	HLA	DQw1	HLA phenotyping
De la Concha EG, et al., 1998 [25]	HLA	DQα1*0101, DQβ1*0501, DQβ1*0503	DNA-typing technique
Verne GN, et al., 1999 [26]	HLA	DQβ1*0602 in white population	DNA-typing technique
Ruiz-de-Leon A, et al., 2002 [27]	HLA	DQα1*0103, DQβ1*0603	DNA-typing technique
Latiano A, et al., 2006 [28]	HLA	DQβ1*0502, DQβ1*0601	DNA-typing technique
Gockel I, et al., 2014 [29]	Immunochip	rs28688207 (HLA-DQβ1)	Genotyping on Illumina Immunochip and TaqMan assay
Becker J, et al., 2016 [30]	HLA	rs28688207	Genotyping by TaqMan Assay
Vackova Z, et al., 2019 [32]	HLA	rs28688207	Genotyping by TaqMan Assay
Furuzawa-Carballeda J, et al., 2018 [33]	HLA	DRβ1*14:54, DQβ1*05:03	Typing by Sanger sequencing and NGS by Illumina
de la Concha EG, et al., 2000 [34]	HLA—TNF	DQ1—TNFα11	Typing by dot-blot hybridization and semiautomatic sequencing method
Wouters MM, et al., 2014 [35]	GWAS	rs1799724 (LTA -TNFα)	Genotyping by Illumina Golden Gate and Sequenom MassARRAY
de León AR, et al., 2010 [36]	IL23R—HLA	rs11209026 (IL23R)	Typing by hybridization and Genotyping by TaqMan Assay
Nuñez C, et al., 2011 [37]	IL10	rs1800896, rs1800871, rs1800872	Typing by hybridization and Genotyping by TaqMan Assay
Latiano A, et al., 2014 [38]	IL33	rs3939286	Genotyping by TaqMan Assay
Santiango JL, et al., 2007 [39]	PTPN22	rs2476601	Genotyping by TaqMan Assay and Typing by hybridization
Singh R, et al., 2015 [40]	eNOS	27-bp VNTR (eNOS), rs1060826 (iNOS), rs2682826 (nNOS)	Electrophoresis and restriction enzyme digestion
Sarnelli G, et al., 2017 [41]	NOS2	10 and 13 CCTTT repeats	Capillary electrophoresis
Paladini F, et al., 2009 [42]	VIPR1	rs437876	Restriction enzyme digestion and SNaPshot technique
Alahdab YO, et al., 2012 [43]	c-kit	rs2237025 and rs6554199	Genotyping by real-time PCR
Santiago JL, et al., 2012 [44]	c-kit	rs6554199	Genotyping by TaqMan Assay
Li Q, et al., 2021 [45]	GWAS -WES	rs1705003 (CUTA), rs1126511 (HLA-DPB1), NC_000007.13:g.28848865G > T (CREB5), NC_000003.11:g.138183253C > T (ESYT3), NC_000002.11:g.11925128A > G (LPIN1)	Array-based GWAS by Illumina and WES by Agilent

HLA: Human Leukocyte Antigen, GWAS: Genome-Wide Association Studies, IL23R: Interleukin-23 receptor, IL10: Interleukin 10, IL33: Interleukin 33, LTA: lymphotoxin-α, PCR: Polymerase Chain Reaction, TNFα: tumour necrosis factor-α, PTPN22: Protein tyrosine phosphatase non-receptor type 22, NOS: Nitric oxide synthase, VNTR: variable number of tandem repeats, eNOS: endothelial Nitric Oxide Synthase, iNOS: inducible Nitric Oxide Synthase, nNOS: neuronal Nitric Oxide Synthase, VIPR1: Vasoactive intestinal peptide receptor 1, c-Kit: Tyrosine-protein kinase, NSG: Next Generation Sequencing, WES: Whole Exome Sequencing, CUTA: CutA Divalent Cation Tolerance Homolog, HLA-DPB1: Major Histocompatibility Complex, Class II, DP Beta 1, CREB5: CAMP Responsive Element Binding Protein 5, ESYT3: Extended Synaptotagmin 3, LPIN1: Lipin 1.

To highlight new signaling pathways involved in neuronal changes, several studies have been conducted. Firstly, Singh et al. demonstrated that different genetic variants of the 3 isoforms that synthesize nitric oxide (NOS) (endothelial NOS-eNOS, inducible NOS-iNOS and neuronal NOS-nNOS), the most important known inhibitory neurotransmitter of the esophageal myenteric plexus, could represent risk factors for the achalasia development. They showed that eNOS4a4a, iNOS22GA and nNOS29TT genotypes were more frequent in patients with achalasia than in healthy subjects [[40], Tables 2–5]. Next, the correlation between the polymorphic pentanucleotide (CCTTT)n in the promoter of the *NOS2* gene and achalasia was evaluated in another study, obtaining an association with the disease likely through an allele-specific modulation of NO synthesis. Furthermore, no significant difference was observed between males and females at the age of diagnosis, nor did gender influence the duration of the disease [41]. Vasoactive intestinal peptide (VIP) is another neurotransmitter involved in LES relaxation. Paladini et al. conducted a study on the variation of 5 SNPs in the human VIP receptor 1 (*VIPR1*) gene in 104 patients with achalasia and 300 controls showing significant differences in the allelic, genotypic and phenotypic distributions of the rs437876 SNP, with a particularly strong association in patients with late onset of the disease [[42], Table 1]. A pilot study of Alahdab et al. focused on the analysis of two functional SNPs (rs2237025 and rs6554199) within the c-kit gene, which encodes a tyrosine kinase receptor expressed by interstitial cells of Cajal (ICCs), that have been shown to be involved in nitrergic neurotransmission of the LES. They found that the frequency of the minor T allele of rs6554199 was significantly higher in Turkish patients with achalasia compared to controls [43]. The association was not confirmed in a Spanish study involving a larger number of patients and controls [44].

Today, next-generation sequencing technologies (NGS), a powerful tool for analyzing DNA and RNA molecules in a high-throughput, have enabled the identification of novel risk genes with low-frequency variants. With this intent, Li et al. performed a sequencing-based exome-wide association study of achalasia in a Chinese population identifying and validating two independent common variants at the HLA region, rs1705003 cuta divalent cation tolerance homolog (*CUTA*) and rs1126511 (*HLA-DPB1*), and three rare and functional variants in cyclic AMP responsive element binding protein 5 (*CREB5*), extended synaptotagmin 3 (*ESYT3*) and lipin 1 (*LPIN1*) genes associated with an increased risk to develop idiopathic achalasia. Interestingly, these rare variants are located in immunological and neurological genes thus reinforcing their involvement in the etiology of achalasia [[45], Tables 1 and 2].

## 5. Transcriptomics

Being an omics science, transcriptomics, as one functional output of genetic regulation, addresses the analysis of RNAs collectively, taking into consideration the cellular RNA pool. Over the past decade, transcriptome analysis has made great progress in identifying risk molecular biomarkers and new therapeutic targets, and in pinpointing key molecular pathways.

Transcriptomics studies performed on achalasia attempted to decipher the altered molecular pathways in patients.

With this purpose in 2016 Palmieri et al. performed the first genome-wide expression profiling of mRNA extracted from the LES muscle specimens of patients with achalasia compared with those of controls, by means of microarray technology and advanced in-silico functional analyses. This approach allowed the identification of 1728 differentially expressed genes and, in particular, genes involved in the smooth muscle contraction biological function, like ephrin receptor A7 (*EPHA7*), tropomyosin 2 (*TPM2*), and integrin alpha 1 (*ITGA*)*,* were the most up-regulated. Interestingly, among the most down-regulated genes, the chemokine (*C-X-C motif*) ligand 17 (*CXCL17*) and the immunoglobulin heavy constant alpha 1 and 2 (*IGHA1-2*) genes, play an important role in the innate defense against infections, supporting the hypothesis of process in triggering achalasia [[46], Table 2, Figure 1]. In another study, quantitative transcriptome and cluster analysis revealed 111 differentially expressed genes in total RNA extracted from esophageal biopsies of achalasia patients and controls, identifying a down-regulation of the genes cysteine rich angiogenic inducer 61 (*CYR61*), connective tissue growth factor (*CTGF*), *c-kit*, dual specificity phosphatase 5 (*DUSP5*), early growth response 1 (*EGR1*) and up-regulation of a-kinase anchoring protein 6 (*AKAP6*) and inositol polyphosphate-4-phosphatase (*INPP4B*) in patients. The *c-kit* downregulation and *INPP4B* upregulation, validated by western blot and immunohistochemistry, extend previous data showing a loss of the ICC network in tissues from patients with achalasia and indicate a general increase in inhibitory signals for cell survival [47]. Furthermore, Patel’s et al. investigation identified, through gene expression analysis, 65 and 120 genes differentially expressed in the distal and in the proximal part of the esophagus of patients with achalasia, respectively, highlighting the importance of better understanding the molecular and regional differences in the disease. In addition, they observed changes in the cellular response induced by cytokine stimulus in distal esophageal biopsies, in line with our knowledge of achalasia and inflammation [[48], Figures 2 and 3].

Other essential component of transcriptomics are the microRNAs (miRNAs) also identified as promising and potential markers for early diagnosis and as therapeutic targets of treatments. miRNAs are short noncoding RNAs of about 18–24 nucleotides, which regulate the post-transcriptional gene expression by RNA silencing contributing to the abnormal expression of pathogenic genes in diseases [49]. There is little information regarding the expression of miRNA in achalasia although the research field is expanding particularly for their association with tumorigenic processes, including cell proliferation, apoptosis, angiogenesis and invasion through their interaction with oncogenes and anti-oncogenes [50]. The first report was by Shoji et al., in order to identify miRNAs specific to the esophageal mucosa of achalasia and to assess their alteration following POEM, analyzed eight patients and four controls using a microarray-based technology, identifying the miR-130a significantly higher in cases. Significant correlation with male and smoking history in patients with achalasia was observed, and no significant change in miR-130a expression before and after the POEM [51]. With the same intent, another study carried out a microarray analysis and subsequent quantitative real-time PCR, finding that two herpes simplex virus-(HSV-1)-derived viral miRNAs, hsv1-miR-H1-3p and hsv1-miR-H18, were significantly overexpressed in muscular samples obtained from LES during POEM from six Japanese patients with achalasia compared to four controls. On the other hand, no significant associations between the expressions of the identified viral miRNAs and morphological types of achalasia and the dilatation grading, and disease duration, were observed [[52], Figure 3]. In the investigation of Palmieri et al., miRNA microarray technology followed by integrated bioinformatics analysis was performed to identify miRNAs with altered expression in 11 tissue specimens of esophageal mucosa of achalasia patients and 5 controls [[53], Table 1 and Figure 6]. Moreover, the interaction networks between significant miRNAs and their target genes revealed miRNAs-mRNAs interacting pairs with genes involved in immune cell trafficking, skeletal and muscular system development, and nervous system development macro-processes. The researchers observed that the down-expression of miR-200c-3p correlated with an increased expression of protein kinase CGMP-dependent 1 (*PRKG1*), sulfatase 1 (*SULF1*), and synapse defective rho GTPase homolog 1 (*SYDE1*) genes, involved in smooth muscle contractility and synaptic transmission of cells pathways. Subsequently, the biological interaction between these genes and miR-200c-3p was explored by employing a combination of computational and molecular analysis resulting in miR-200c-3p regulated directly *PRKG1* gene and indirectly *SULF1* and *SYDE1* genes, influencing the pathogenesis of achalasia through NO/cGMP/PKG signaling [54].

The latest NGS technologies have been also successfully applied to the analysis of the transcriptome. By this approach, miRNAs associated with the development of achalasia were identified. A case-control study was performed on the esophageal tissue samples from 52 patients with achalasia and 50 controls. Fifteen miRNAs were significantly differentially expressed with predicted targets significantly related to neurotransmission, axon development and regeneration, cellular response to nerve growth factors and inflammation processes. Moreover, further analysis using quantitative real-time PCR showed significant down-regulation of miR-217 in the LES samples of the achalasia patients with significant enrichment in myelination process ontology [55]. Recently, Liu et al. have introduced the use of the single-cell RNA sequencing technique (scRNA-seq), an unbiased approach for characterizing cell diversity and heterogeneous phenotypes at high resolution, to outline a gene profile of immune cells present in the blood and paired LES tissues of patients affected by achalasia compared with healthy controls. The results revealed the presence of C1QC^+^ macrophages and resident memory T cells with a specific composition and transcriptional phenotype, localized surrounding or even infiltrating the myenteric plexus in the LES of the patients and more activated in type I compared with type II achalasia, suggesting their involvement in the inflammatory process and pathophysiology of the disease [56].

## 6. Metagenomics

Metagenomics is not only one of the latest systems omics science technologies but also one that has arguably the broadest set of applications and impacts globally. It is the study of the structure and function of entire nucleotide sequences from a community of organisms in a given environment. Recently, precise evaluations of microbial communities in the human body have become possible through technological advancements made by high-throughput microbial sequencing techniques.

Although underinvestigated, the evaluation, the composition and role of the esophageal microbiota in achalasia and the exploration of the potential microbial mechanisms involved in its pathogenesis have been considered [57].

In 2003, Pajecki and colleagues conducted a pioneering qualitative and quantitative analysis analyzing the esophageal microbiota in 15 patients with Chagas’ megaesophageal disease (a complication of achalasia) compared with 10 subjects with a normal esophagus. They observed that in the patients there was great variability in bacteria, constituted mainly of aerobic Gram-positive (*Streptococcus*) and anaerobic Gram-negative (*Veillonella*) bacteria, in concentrations correlated to the degree of esophageal dilation, with higher levels in the more advanced phases of megaesophagus, where the dilation and therefore the stasis are more evident [58]. With the hypothesis that the esophageal microbiota can be altered by long-term stasis of food and oral bacteria, resulting in chronic inflammation of the esophageal mucosa and increased risk of esophageal squamous cell carcinoma (ESCC), a study was carried out through 16S rRNA gene sequencing analysis. The authors compared the oral microbiota, obtained using a swab, and the esophageal one using a brush via endoscopy, of 6 patients with achalasia before and after POEM, and 14 with ESCC. The results indicated that in patients with achalasia, the buccal and the esophageal microbiota were significantly different with the *Streptococcus* being the most abundant in both microbiota, did not change after POEM, and was not different from those with ESCC [[59], Figure 3]. This concept was also addressed by Jung et al. who investigated the composition of the microbial community of the 29 Korean patients with achalasia before and 8 weeks after POEM, only in the esophageal samples (mucosal biopsies and retention fluid). They showed that *Streptococcus* was the most abundant at the genus level, and the *Firmicutes*, *Bacteroidetes*, *Proteobacteria*, *Actinobacteria* and *Fusobacteria* were the dominant phyla with significant differences between the retention fluid and the tissue, and no significant changes post-POEM were observed [60]. Subsequently, through a prospective case-control study, Yeh et al. used the NGS approach to evaluate the characteristics of the esophageal microbiota, collected by endoscopic brushing, in 31 achalasia patients and 15 asymptomatic subjects, and its changes before and 3 months after POEM. The study reconfirmed the presence of a different composition of the esophageal microbial community in patients with achalasia compared to the control group, with an increase in *Firmicutes*, *Bacteroidetes* and *Proteobacteria* at the phylum level, and in *Streptococcus*, *Prevotella*, *Lactobacillus*, *Veillonella*, *Neisseria* and *Alloprevotella* at the genus level. The discriminating enriched genera in patients was *Lactobacillus*, which was also associated with achalasia severity. In addition, after POEM treatment, an increase in *Neisseria* and a decrease in *Lactobacillus* was observed with a high prevalence of erosive esophagitis and an increase in *Lachnoanaerobaculu* genus [[61], Figure 3]. Similarly, in another study, mucosal biopsies were collected from 32 achalasia patients before POEM, and 27 healthy individuals were analyzed using 16S rRNA sequencing. Clear differentiation between samples was observed, with the *Novosphingobium*, *Aquabacterium* and *Lactobacillus* genera being significantly enriched in patients, and *Streptococcus*, *Acinetobacter* and *Pseudomonas* significantly more abundant in controls. Furthermore, they used animal models to validate achalasia by inducing esophageal dysbiosis in mice via chronic exposure to ampicillin sodium. The dominant phyla in the esophageal microbiota of both control and antibiotic-treated mice were *Firmicutes*, *Proteobacteria*, *Actinobacteria* and *Bacteroidetes*. In mice that received antibiotics, the relative abundance of *Proteobacteria*, *Bacteroidetes* and *Actinobacteria* significantly increased, while that of *Firmicutes* decreased, compared to the control group, suggesting that long-term antibiotic exposure leads to structural and histological changes in the esophagus similar to those observed in patients with achalasia. In addition, these mice exhibit decreased myenteric neurons, impaired esophageal function, increased macrophages in the lamina propria, up-regulated TLR4-MYD88-NF-κB pathway, and elevated production of pro-inflammatory cytokines TNF-α, IL-1β, and IL-6 [[62], Figures 1–6]. Lastly, a recent paper by Ikeda et al. analyzed the microbiota of esophageal mucosa and LES samples from patients with type II achalasia who underwent POEM, and transferred the esophageal conditioned media obtained from patients into the esophagus of mice, with the aim to study the alterations in esophageal smooth muscle contraction and the associated inflammatory response. The authors reported an increase in the α-diversity index of the microbiota and of specific microbes, such as *Actinomyces* and *Dialister*, with a positive correlation between the levels of *Actinomyces* and IL-23A, and between *Dialister* and IL-17A, IL-17F and IL-22 levels. Moreover, in LES of patients with achalasia, the hypophosphorylation of 20-kDa myosin light chains, a possible cause of impaired contractility, was associated with the myosin phosphatase-inhibitor protein CPI-17 down-regulation and with an increase in Th17-related cytokines, including IL-17A, IL-17F, IL-22 and IL-23A [63].

## 7. Viromics

The virome is the viral fraction of the microbiome. In recent years, with the development of high-throughput nucleic acid sequencing technology that has facilitated the identification and characterization of viruses in the human virome, also this field is constantly expanding with attractive and promising pathophysiological implications.

Several articles have attempted to explore evidence supporting a viral etiology for achalasia as a potential triggering agent of the disease, albeit with controversial results.

The first investigation was conducted by Robertson et al. in 1993 with the aim of searching for past or present infection with herpes viruses, serum antibody titers to herpes simplex virus-type 1 (HSV-1), cytomegalovirus (CMV) and varicella zoster virus (VZV), by complement fixation test, in 58 patients with achalasia and 40 controls without esophageal disorders. In situ hybridization was subsequently applied on resected esophageal tissues from 9 patients with achalasia and 20 controls. Low antibody levels to all three viruses were detected, with only a significant increase in antibodies against VZV in the achalasia patients compared to controls, and three patients showed evidence only of VZV virus in tissues containing myenteric plexus [64]. Contrariwise, no sequences of herpesviruses (HSV-1 and 2, CMV, Epstein-Bar virus (EBV), VZV, and human herpesvirus 6 (HHV-6)), measles, and human papilloma virus were detected in myotomy specimens from 13 achalasia patients, 9 esophageal cancer patients, and 6 fetuses autopsy specimens, with PCR technique [65]. The serologic study, using enzyme-linked immunosorbent assay (ELISA), conducted on 15 achalasia patients and 8 controls to determine the presence of specific anti-HSV-1 and anti-HSV-2 antibodies, revealed no significant differences between the two groups. Although an increase in the proliferative index in mononuclear cells from achalasia patients stimulated with HSV-1 in comparison with control subjects was observed, no HSV-1 DNA was detected by PCR in the esophageal muscle samples [66]. Next, Facco et al. demonstrated, by flow cytometry evaluation and CDR3 length spectratyping analysis, that the LES of achalasia patients was characterized by a significantly higher lymphocytic infiltrate than controls, mainly represented by CD3+CD8+ T cells. The characterization of the T-cell receptor beta chain repertoire of CD3+ cells showed the expression of a limited number of T-cell receptor beta variable gene families, suggesting a disease-associated oligoclonal selection of T cells. Furthermore, a strong reactivity toward HSV-1 antigens by lymphocytes from achalasia LES was observed as showed by increased proliferation and Th-1 type cytokines release [[67], Table 2]. On the other hand, Lau et al. demonstrated that peripheral blood mononuclear cells (PBMCs) of patients with achalasia showed an enhanced systemic immune response to HSV-1 antigens compared to controls proved by an increase in interferon-γ production [68]. Afterward, from the evaluation of the myenteric plexus of 12 achalasia patients undergoing esophagomyotomy compared with esophageal tissue from 7 controls for the possible presence of herpesvirus, assessed by immunohistochemistry, and human papillomavirus (HPV), by in-situ hybridization, no evidence for the presence of the viruses investigated emerged. Moreover, a significant decrease in both enteric neurons and ICC, the absence of immunoreactivity for VIP and myenteric plexitis was displayed [69]. A case-control analysis was published by Naik et al. in Gastroenterology 2021 where the authors demonstrated their hypothesis that the reactivation of VZV from latency in esophageal neurons gives rise to chronic VZV infection that impairs the functional regulation of esophageal motility of the LES in achalasia. Results indicated that 80% of the achalasia patients analyzed (*n* = 15) had detectable salivary VZV DNA, and 87% had transcripts encoding VZV late gene products in esophageal tissue. In addition, the immunoreactivities of VZV late proteins were detected in enteric neuronal cell bodies and nerve fibers within the esophageal wall consistent with the concept that infected esophageal neurons produce virions that enter axons [70]. Conversely, the study by Moradi et al. failed to detect any significant association between achalasia patients and the genomes of neurotropic DNA viruses (CMV, VZV, EBV, John Cunningham virus (JCV), HSV1, HHV-6), neurotropic RNA viruses (Bornavirus, Coxsackie virus, measles, Human T-Lymphotropic Viruses 1 and 2 (HTLV-1, 2)) or non-neurotropic viruses (adenovirus and HPV) [71]. Hitherto, regards the possible association between the development of achalasia and the severe acute respiratory syndrome coronavirus 2 (SARS-CoV-2 virus), there are only two reports in the literature [[72,73] Table 1]. The authors focused on the evaluation of the presence of the SARS-CoV-2 virus in the muscle tissue of patients with and without post-COVID-19 achalasia. Although these findings cannot definitively affirm that SARS-CoV-2 is causative for achalasia, the rapid development of the disease in asymptomatic patients and the presence of SARS-CoV-2 in the myenteric plexus of achalasia patients who had experienced COVID-19 leave open this possibility.

## 8. Proteomics

Proteomics is the large-scale study of proteomes, a set of proteins produced in an organism, system, or biological context at a precise developmental or cellular phase. With the development of experimental technology, the proteomics techniques evolved from conventional methods such as immunohistochemistry staining, western blot and ELISA, up to high-throughput methods such as tissue microarray, protein pathway array and mass spectrometry, hold great promise for uncovering the molecular mechanisms that underlie diseases [74].

Based on the proposed role of an inflammatory component mediated by the immune system and ganglia degeneration in the etiology of achalasia, few reports evaluating specific protein alterations have been proposed (Table 2).

Wang et al. using traditional ELISA laboratory methodology, found increased concentrations of the IL-17 and IL-22 in the serum of 14 patients with achalasia who underwent POEM compared with 14 healthy individuals. Furthermore, immunohistochemistry analysis performed on LES myofilaments revealed significant differences between the groups with expression of IL-17 and IL-22 mainly in cytoplasm and nucleus [75]. Another study compared the plasma immunological profiles of TNF-α, IL-6, IFN-γ, IL-12, IL-17, IL-22 and IL-23 in 53 patients with achalasia, 22 eosinophilic esophagitis (EoE), and 20 gastroesophageal reflux disease, through Quantikine Human Immunoassays. Statistically significant increases in IL-6 in patients with achalasia vs. EoE patients were observed, and no difference in cytokines levels between the three subtypes of achalasia was demonstrated [76]. Similarly, the study by Chen et al. evaluated the levels of serological cytokines and chemokines in 47 patients with achalasia and 47 matched healthy people analyzed by Luminex xMAP immunoassay (Cytokine 48-plex and TGFβ 3-Plex). Patients with achalasia exhibited increased concentrations of eleven cytokines and chemokines, namely TGF-ß1, TGF-ß2, TGF-ß3, IL-1ra, IL-17, IL-18, IFN-γ, MIG, PDGF-BB, IP-10 and SCGF-B. Moreover, twelve biomarkers were significantly increased in type III compared with I and II achalasia, namely, TGF-ß2, IL-1ra, IL-2Ra, IL-18, MIG, IFN-γ, SDF-1a, Eotaxin, PDGF-BB, IP-10, MCP-1 and TRAIL [[77], Tables 3 and 4]. Next, bioinformatics analysis revealed that the regulation of signaling receptor activity and receptor-ligand activity were the most related pathways of these cytokines and chemokines. Subsequently, Bio-Plex Pro™ Human Cytokine 27-plex assay kit was applied to measure the concentration of a panel of cytokines in plasma collected from 12 patients with achalasia before and after POEM and 10 healthy subjects. The levels of IL-17, IL-1β, C-C motif chemokine ligand 2, IL-4, IL-5, IL-1ra, IL-7, IL-12, interferon-γ, IL-2, fibroblast growth factor-2, colony-stimulating factor (CSF)2 and CSF3 were significantly higher in patients. However, no significant differences in cytokine production before and after POEM were observed [78]. On the other hand, serum protein levels of all the markers, tested with the same assay kit in 68 patients with achalasia and 39 controls, reached no statistical significance between the two groups analyzed [79]. An alternative method for the proteomics approach was applied by Im et al. They used matrix-assisted laser desorption ionization/time-of-flight mass spectrometry (MALDI-TOF/MS) analysis to compare the protein profiles of 5 patients with achalasia and 5 healthy individuals, identifying 28 protein spots up-regulated in patients, including complement C4B5, C5 and C3, cyclin-dependent kinase 5, transthyretin and alpha 2 macroglobulin. Additionally, by ELISA, higher serum C3 levels in achalasia patients with respect to controls were detected [[80], Table 2]. These findings highlighted key proteins that suggest an immune-mediated response or neuronal degeneration in the cause of achalasia. Recently, another study, using liquid chromatography-tandem mass spectrometry (LC–MS/MS), focused on the identification of differentially expressed proteins and potential pathways across paired LES muscle and serum samples from 24 achalasia subtypes and 20 controls. Distinct proteomics patterns of serum and muscle samples between achalasia patients and controls were shown, and functional enrichment analysis revealed that differentially expressed proteins were involved in immunity, inflammation, infection and neurodegeneration pathways. Moreover, proteins involved in the extracellular matrix-receptor interaction increased sequentially between the control group, type III, type II and type I achalasia [[81], Table 2 and Figure 2].

## 9. Summary and Future Directions

Achalasia is a condition affecting the motility of the esophagus, its origin is not well understood, diagnosis can be complex, and there is no definitive cure. Despite significant advances in the diagnosis and treatment, many aspects are still unclear, and future developments in the management of the disease are needed. Promising directions concern the improvement of surgical techniques and pharmacological therapies with the development of new treatment modalities that are increasingly less invasive and longer lasting, as well as the development of regenerative therapies capable of restoring normal esophageal function.

Achalasia is characterized by a loss of inhibitory ganglia and neurons in the esophageal neural plex. Although the cause of this neuronal degeneration is mainly unknown, immune cell- and antibody-mediated mechanisms seem to be involved, possibly triggered by infectious agents, in individuals with a genetic susceptibility [3].

In order to gain a fuller understanding of achalasia, some studies have been conducted to identify achalasia-associated molecular traits from genomics, transcriptomics, metagenomics, viromics and proteomics levels, but reports on achalasia multi-omics are missing, so far.

What emerges from this study of literature review on achalasia omics is that the field of genomics has allowed the identification of rare genetic variants and polymorphisms in likely candidates genes encoding for proteins involved in the immune response. Genetic risk factors within the *HLA-DQ receptor*, SNPs on the *IL23R* which regulates T cell differentiation, on *PTPN22*, which is a down-regulator of T-cell activation, and *IL10* gene promoter were identified, emphasizing the concept of a genetically driven immune response in patients with achalasia. It is noteworthy that the large number of identified variants located in the HLA region highlights, given the prevalence of achalasia, the role of common genetic variants in rare diseases such as achalasia, and emphasizes the importance of immune-mediated processes in its etiology [29,45]. Thus, the common variant rs1705003, may affect the expression of *CUTA*, a copper-related protein, and trigger a cascade of neurodegenerative steps [82] which may contribute to the genetic basis of the neurodegenerative process and result in loss of esophageal peristalsis [45]. Moreover, functional studies are now required to demonstrate the exact cellular effect of the identified risk variants and the alterations they could lead to on immune-mediated processes in the disease. Deranged esophageal peristalsis and loss of LES function, due to imbalance between excitatory neurons releasing acetylcholine and inhibitory neurons that release NO and VIP, are the abnormalities and diagnostic characteristics of achalasia [83]. NO is thought to mediate the esophageal motor function through his concentration [84], and aberrant iNOS expression may have detrimental consequences, as excessive NO production has been proved to exert neurotoxic effects [85]. nNOS is known to be the main source of NO in the LES and is involved in gastrointestinal abnormalities seen in achalasia [86]. VIP is also an inhibitory neurotransmitter that plays a similar role to NOS based on its involvement in smooth muscle relaxation in the esophageal myenteric plexus. In addition, it may play a role in the regulation of inflammation [87]. Moreover, recent evidence suggests that ICCs, a group of specialized cells that constitutively express c-kit, are involved in NO-mediated inhibitory neuromuscular transmission [88]. Thus, polymorphisms in *NOS*, *VIP* and *c-kit* genes represented ideal candidates to explain the inflammation and the inhibitory neurotransmission observed in achalasia [89]. With the advent of WES technology, more and more rare variants involved in the development of achalasia will be investigated, and an improved functional evaluation may help increase the validation rate in the study of rare variants. Three rare and functional variants were successfully found to contribute to the risk of disease and tended to be located in genes (*CREB5*, *ESYT3*, *LPIN1*) affecting nerve and muscle function, and co-occur in achalasia-affected individuals with a combined effect, providing new insight into the etiology of the disease [45]. Genomics studies of achalasia, albeit emphasizing the genetic predisposition underlying the mechanisms of achalasia, have identified a series of common loci and rare genetic variants associated that only underline the susceptibility and increased risk of development of the disease without generating clear or definitive evidence of its etiology, pathogenesis, diagnosis, treatment or therapy. The small sample size, lack of reproducibility and validation in large data sets, and the unreliability of the clinical phenotype represent the main limitations of the reported polymorphisms in identifying a linking factor in the pathophysiology. Indeed, no clinically useful achalasia genetic markers have been identified so far, and there is still no evidence that assessing genetic factors independently of other contributing factors can provide a practical guide for diagnosis or therapy. In addition, being complex diseases, its etiology is multifactorial and it could be that a combination of the cumulative effect of variants in various risk genes and environmental factors leads to the disorder.

Transcriptomics studies have revealed alterations in gene expression helping to understand the mechanisms underlying the pathology. An up-regulated series of mostly differentially expressed genes involved in smooth muscle contraction and in innate defense against infections where identified, supporting the hypothesis of a viral process in triggering achalasia. Pathway enrichment analysis reported the neuronal/muscular/immunity organismal responses as critical physiological categories of processes, and several categories related to the contractility of muscle, damage to the nervous system, synaptic transmission activity, cytokine response and viral defense were also observed supporting the concept that degenerative alterations, primarily evoked by a chronic inflammatory infiltrate targeting the myenteric plexus, lead to achalasia [46,48]. Interestingly, changes in gene expression levels and differentially regulated pathways were highlighted in the proximal and distal esophagus of type I versus type II achalasia patients, corroborating the hypothesis that type I is believed to be a more advanced stage of achalasia compared to type II [90], and the concept that at molecular level type I achalasia represents a progression of the disease [48]. These findings suggest that including separate analyses of the proximal and distal esophageal mucosa could offer additional insights into the disease. Different expression levels of achalasia-associated miRNAs, and potentially regulated genes were identified. Although dysregulated miRNAs such as miR-130a [51], HSV-1-derived viral miRNAs [52], miR-133b, miR-133a-3p, miR-375 and miR-200c-3p [53,54], miR-217, miR-143-3p and miR-133a-5p [55] were pointed out, their role in achalasia is not yet fully clarified. The results obtained are conflicting, and different sets of miRNAs were identified from each study. Thus, further studies, replication with larger numbers of study participants and patients, and validation are needed to assess the utility of miRNAs as potential biomarkers in vivo and their application in the diagnosis and probably treatment of the disease.

Metagenomics, analyzing the microbial communities of the esophagus and LES, has revealed alterations in the microbiota of achalasia patients that may influence esophageal inflammation and neuromuscular function. The importance of the esophageal microbiota to achalasia pathogenesis has gained increasing attention in recent years. It comes from the hypothesis that the esophageal microbiota can be altered by long-term stasis of food and oral bacteria, creating a favorable environment for microbial growth, resulting in inflammation of the esophageal mucosa. Furthermore, the esophageal microbiota is influenced by diet, age and modulated according to the degree of esophageal dilation, which is greater in the advanced stage of the disease and in response to treatment, suggesting the existence of a dynamic bacterial background. Pilot investigations revealed that the oral and esophageal microbiota were significantly different in patients with achalasia, and some of the composition changed after POEM [59,61]. It is characterized by reduced microbial diversity, decreased abundance of Gram-positive bacteria and higher representation of Gram-negative bacteria which may induce aberrant esophageal motility [62]. Studies are progressively increasing, but have generated unclear and inconsistent results so far, basically focused on the microbiota composition before and after POEM. Further studies using advanced molecular and analysis techniques will be helpful to better understand the microbiota composition and diversity of patients with achalasia and the exact mechanisms by which microbiota contributes to the disorder. Interestingly, by targeting a microbiota disease-specific signature, it may be possible to modulate and improve the dysbiotic microenvironment of the esophagus to relieve stasis in the esophageal lumen at the time of diagnosis or as a treatment for achalasia

Viromics has pointed out the presence of specific viruses in patients with achalasia highlighting a possible role of viral infections in the onset and development of the disease. The concept that achalasia may be the result of an immune-mediated inflammatory disorder triggered by a viral infection has been widely debated [91]. Several proposed candidate viruses such as HSV, VZV, CMV, HPV and recently SARS-CoV-2 [72,73], have been hypothesized to play a causal role in triggering tissue inflammation. Chronic viral infection may elicit an aberrant immune response and, under appropriate genetic and environmental settings, the loss of inhibitory neurons in the myenteric plexus of the LES, resulting in failure of muscle relaxation and deranged esophageal peristalsis [92]. Viral DNA/RNA and virus-targeted antibodies were found in esophageal tissue, serum and saliva of patients with achalasia with conflicting results among studies, no clear causal relationship has yet been established, and their role in achalasia pathogenesis remains largely unknown. Since not all infected patients develop the disease, the potential role of genetic factors to make some individuals more susceptible to achalasia than others has been suggested [91]. Recent evidence suggests the hypothesis that VZV plays a causal role in the disease since its reactivation from latency in esophageal neurons gives rise to chronic VZV infection that impairs the functional regulation of esophageal motility, and LES control in achalasia [70]. Again, further confirmation of causality is needed to better decipher the process that leads to the possible neuronal reactivation of viruses and to uncover additional factors, possibly genetic, that may be involved. Furthermore, in addition to the relationship between viral infections and the onset of the disorder, a definite cause-effect relationship should also be explored. Finally, there is still no evidence suggesting that vaccination against these viruses plays a protective role in the development of achalasia, as it could be a useful means of improving the management of patients suffering from it. So far, research has focused on the microbiota of achalasia patients, while studies conducted on the viral community (virome) remain uncharacterized. The interaction between viruses with both bacteria and host cells, which influences overall intestinal and esophageal homeostasis [93], could be the basis for new research on the contribution of populations of viruses to disease pathogenesis.

Proteomics has identified specific altered protein profiles in the patients offering new insights into pathological processes. Immune-mediated neuronal and ganglia degeneration has also been proposed as an important etiology of achalasia [94]. Most likely, the myenteric neurons disappear because of chronic ganglionitis as proved by a substantial immune cell infiltration of cytotoxic lymphocytes eosinophils, and mast cells expressing activation markers [95]. Characterization of infiltrating cells in resection specimens on patients with achalasia has largely been performed using histological or immunohistochemical analyses, with conflicting results [96]. Conversely, investigations evaluating specific protein alterations and the involvement of circulating levels of inflammatory mediators, cytokines, chemokines and growth factors in patients with achalasia, remain insufficient. Different levels of cytokines, chemokines and growth factors were detected in the serum and plasma of achalasia patients compared to healthy individuals [67,68,69] with non-comparable results, suggesting that further research is warranted to identify specific inflammatory biomarkers in patients that contribute to the early diagnosis of the disease and to improve our understanding of the etiopathogenesis of achalasia. Noteworthy, results from the recent high-throughput 4D label-free proteomic study of achalasia indicated that there were specific protein alterations in both the serum and muscle of achalasia, involving immunity, inflammation, infection and neurodegeneration pathways and potential molecular pathways associated with different disease stages [71]. All these findings provide key molecules related to achalasia pathogenesis that might guide the future development of new a suitable treatment. Future efforts at decreasing the myenteric inflammation in its early stages, perhaps by immunosuppressive therapy, could prevent progression to end-stage disease.

Nevertheless, current achalasia studies present limitations. First of all, the quest is at a preliminary stage and the results depend on the number of subjects analyzed and the technology employed. In addition, the majority of reports are still based on simple case-control association studies conducted using different molecular technologies. Thus, to optimize the use of single-omics data for a futuristic integrated multi-omics approach, and for research to provide true advances in the field of achalasia and afford satisfactory contributions to clinical patient management, must consider a number of factors. A multi-omics approach must take into account the complexity of the disease, environmental factors, the modifying factors such as diet, drugs, stress status and sex. In this context, the role of sex, hormonal and immunosuppressive factors that could potentially influence the pathophysiology of achalasia have been considered in the literature, with controversial results [30,36,39,41,51]. No discernible variations in the clinical presentation of achalasia or its medical management have been observed between the sexes [97]. Furthermore, particular caution must be taken in the study design, paying attention to the biosamples to be analyzed (blood, serum, plasma). To ensure the quality and long-term stability of specific biomarkers in biological samples during long-term storage, optimal storage conditions are required to avoid degradation and inactivity of biological molecules, and in the case of tissue, the disease location, stage and type of achalasia, as well as whether the sampling was completed before and after treatment, should also be considered. Another important factor to improve the accuracy and to generate interpretable results is the sample size which should have a higher number of omics data with an average number of samples than a large number of samples with few omics data [98].

In this intricacy, the application and integration of the examined single-omics in the achalasia field, and the right interpretation of results, will offer promising possibilities to deepen the understanding of the molecular mechanisms underlying the physiopathology of the disease and obtain a global and multidimensional view. Recently, the Inflammatory Bowel Disease (IBD) Transcriptome and Metatranscriptome Meta-Analysis web app (IBD TaMMA, https://ibdmeta-analysis.herokuapp.com/, accessed on 18 July 2024) [99], a complete survey of all public data sets generated for IBD-related studies, and EoE-related public data sets visualized in the EoE TaMMA interactive web app (https://eoe-meta-analysis.herokuapp.com/, accessed on 18 July 2024) [100], were developed. The similar computational framework could be created for achalasia by examining the different types of omics data already published, with a multi-omics approach, trying to unravel the molecular landscape of achalasia patients, and resolve its complex pathogenesis, and highlighting specific biomarkers for better patient management and treatment.

Likewise, these multidisciplinary approaches can identify biomarkers to improve screening, stop the disease at early stages, and define new therapeutic targets, allowing us to move closer to precision personalized medicine.

Interestingly, based on satisfactory results regarding the use of Artificial Intelligence in other complex diseases, the adoption of this modern approach could be also considered for achalasia in the future. In addition, the use of wearable devices for real-time monitoring of patients could offer numerous benefits, including the recording of symptoms such as dysphagia, chest pain and reflux, the analysis of eating patterns, immediate feedback on patient behaviors and synchronous transmission of the collected data to the doctor. This information could be valuable for optimizing and improving the quality of life of achalasia patients. Thus, interdisciplinary collaboration between clinicians, researchers and biomedical engineers is hoped and will be crucial to translate these discoveries into practical clinical solutions.

In conclusion, while recent advances in diagnosis and treatment options offer new hope for patients with achalasia, an integrated approach combining omics, technological innovations and molecular discoveries will be essential, although extremely challenging and far from reaching, to further improve the management of this complex disease (Figure 3).

## Figures and Tables

**Figure 1 ijms-25-10148-f001:**
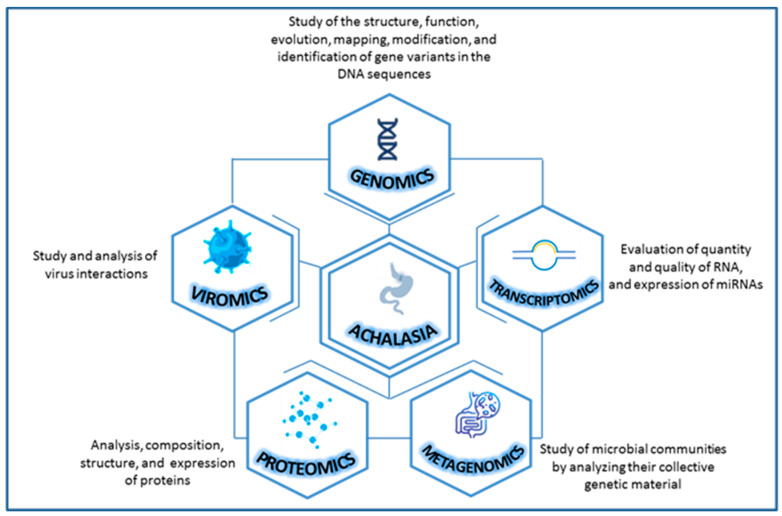
Combining various types of omics data for a more comprehensive study of achalasia.

**Figure 2 ijms-25-10148-f002:**
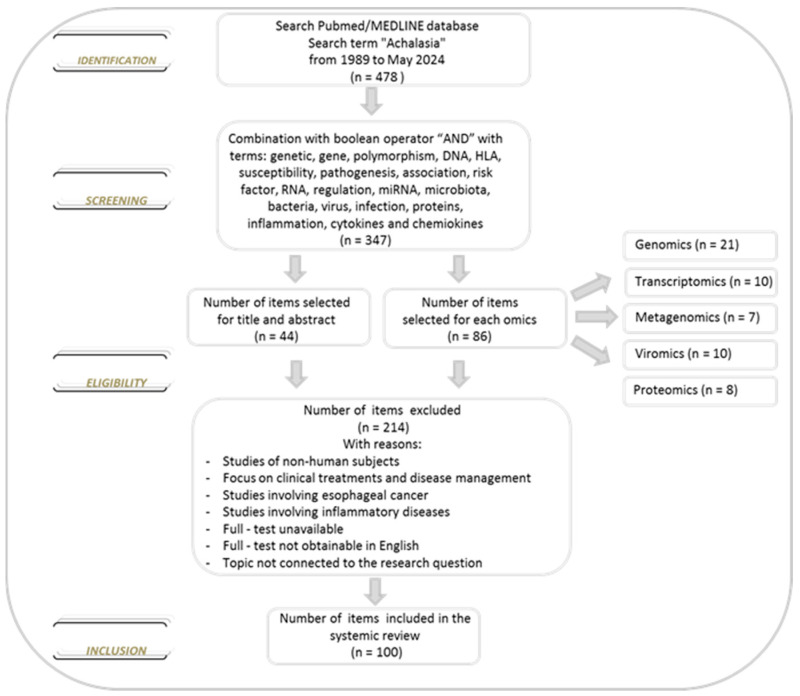
Flowchart of the selection of the literature evaluated in this review.

**Figure 3 ijms-25-10148-f003:**
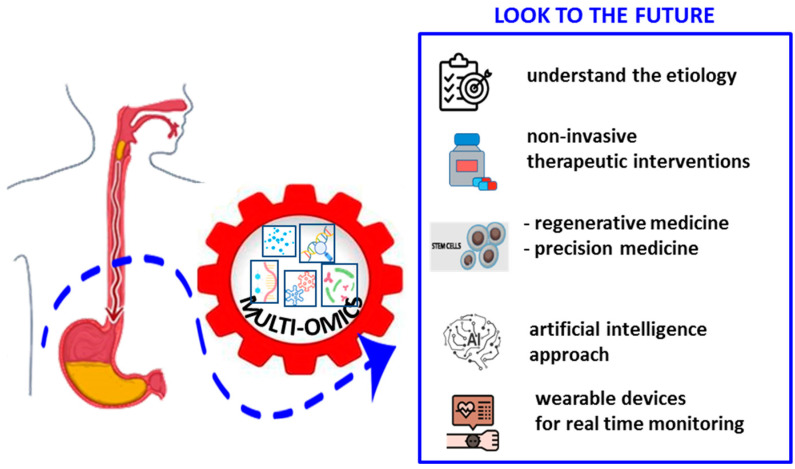
Achalasia data analysis and integration: combination of omics, including genomics, transcriptomics, metagenomics, viromics and proteomics data for possible future developments.

**Table 2 ijms-25-10148-t002:** Comprehensive overview of proteomics studies on achalasia.

Study	Gene	Main Finding	Methods
Wang Z, et al., 2018 [75]	IL-17, IL-22	↑ IL-17, ↑ IL-22	Enzyme-linked immunosorbent assay
Clayton S, et al., 2019 [76]	IL-6, IL-12p70, IL-17, IL-22, IL-23, TNFα and IFNγ	↑ IL-6 (achalasia vs. EoE)	Enzyme-linked immunosorbent assay
Chen WF, et al., 2020 [77]	Cytokine Panel 48-Plex, TGF-β 3-Plex	↑ TGF-ß1, TGF-ß2, TGF-ß3, IL-1ra, IL-17, IL-18, IFN-γ, MIG, IP-10, PDGF-BB, SCGF-B	Immunoassay
Kanda T, et al., 2021 [78]	Cytokine Panel 27-Plex	↑ IL-17, IL-1β, CCL2, IL-4, IL-5, IL-1ra, IL-7, IL-12, IL-2, IFNγ, FGF-2, CSF2, CSF3	Immunoassay
Panza A, et al., 2021 [79]	Cytokine Panel 27-Plex	*ns*	Immunoassay
Im sk, et al., 2013 [80]	Proteins	↑ C4B5, C3, CDK5, TTR, A2M	Matrix Assisted Laser Desorption/Ionisation Time-of-Flight Mass Spectrometry
Chen S, et al., 2023 [81]	Proteins	THYMOSIN β(4), TMOD3, PDCD6IP, Pin1, HMGB2, LEI, TALDO, H1.4,SH3KBP1, Hpr, ALDH16A1, AZU, ARPC2, CALL3, GAL-1, CD34, FKBP1A, LASP-1, INTEGRINA α-IIb, MULTIMERIN-2, HGF, MIF, HISTONE H2A TYPE 2-C, MCP, TC-1	Liquid Chromatography with tandem mass spectrometry

The results reported in the table were obtained comparing patients with achalasia vs. controls except for Clayton’s study. IL-17: Interleukin-17, IL-22: Interleukin-22, IL-6: Interleukin-6, IL-12p70: Interleukin-12p70, Interleukin-17, IL-22: Interleukin-22, IL-23: Interleukin-23, IFNγ: interferon gamma, TNFα: Tumour necrosis factor alpha, TGFβ-1,TGFβ-2,TGFβ-3: transforming growth factor beta-1,2,3, IL-1ra: Interleukin-1, IL-18: Interleukin-18, MIG: monokine induced by gamma, IP-10: interferon gamma-induced protein-10, PDGF-BB: platelet-derived growth factor-BB, SCGF-B: stem cell growth factor-B, IL-1β: Interleukin-1βeta, CCL2: C-C motif chemokine ligand 2, IL-4: Interleukin-4, IL-5: Interleukin-5, IL-1ra: Interleukin *1* Receptor Antagonist, IL-7: Interleukin-7, IL-12: Interleukin-12, IL-2: Interleukin-2, FGF-2: Fibroblast growth factor-2, CSF2: Colony-stimulating factor 2, CSF3: Colony-stimulating factor 3, C4B5: Complement C4B5, C3: Complement C3, CDK5: Cyclin-dependent kinase 5, TTR: Transthyretin, A2M: Alpha 2 macroglobulin, Thymosin β(4): Thymosin Beta-4, TMOD3: Tropomodulin-3, PDCD6IP: Programmed cell Death 6 Interacting Protein, Pin1: Peptidyl Prolyl cis-trans Isomerase NIMA-Interacting 1, HMGB2: High Mobility Group protein B2, LEI: Leukocyte Elastase Inhibitor, TALDO: Transaldolase, H1.4: Histone H1.4, SH3KBP1: SH3 Domain Containing Kinase Binding Protein 1, Hpr: Haptoglobin Related Protein, ALDH16A1: Aldehyde Dehydrogenase 16 family member A1, AZU: Azurocidin, ARPC2: Actin Related Protein 2/3 Complex Subunit 2, CALL3: Calmodulin Like Protein 3, Gal-1: Galectin-1, CD34: Hematopoietic Progenitor Cell Antigen CD34, FKBP1A: Peptidil-prolil cis–trans isomerasi FKBP1A, LASP-1: LIM and SH3 domain protein 1, HGF: Hepatocyte Growth Factor, MIF: Macrophage Migration Inhibitory Factor, MCP: Membrane Cofactor Protein, TC-1: Transcobalamin-1, EoE: eosinophilic esophagitis.

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
