# Peer review of "Focus on Achalasia in the Omics Era"

_ijms, 2024, doi:10.3390/ijms251810148_

Round 1
Reviewer 1 Report
Comments and Suggestions for Authors
For achalasia, the authors reviewed the relevant literature on the etiology and pathogenesis of achalasia using high-throughput analysis of DNA, RNA and biomolecular techniques, and provided an effective method to understand achalasia using multi-omics methods, hoping that multi-omics techniques could provide a basis for clinical application. This is a valuable review. There are a few issues to be aware of here.
1. Figures 2 and 1 are in fact highly repetitive, and it is recommended that the author keep only one.
2. In addition, the authors had elaborated on different omics, and suggested that the authors can cite some important charts in these literatures.
3. It is suggested that the authors can summarize the key genes and proteins related to the diagnosis of achalasia found by different omics in Table Form.
4. How to integrate multi-omics to conduct research on achalasia for the main conclusions?
5. As a clinical application, in fact, it is very concerned which key markers can be used for the diagnosis of achalasia? At the same time, there is a lot of concern about the feasibility of these markers as prognostic analysis, can the authors summarize and discuss the relevant aspects?
Author Response
Reviewer 1
For achalasia, the authors reviewed the relevant literature on the etiology and pathogenesis of achalasia using high-throughput analysis of DNA, RNA and biomolecular techniques, and provided an effective method to understand achalasia using multi-omics methods, hoping that multi-omics techniques could provide a basis for clinical application. This is a valuable review. There are a few issues to be aware of here.
Comments 1: Figures 2 and 1 are in fact highly repetitive, and it is recommended that the author keep only one
Response 1: We thank we reviewer for appreciating our review. With the intent to save both figures, as they depict two different concepts, we modified the figure 2.
Comments 2: In addition, the authors had elaborated on different omics, and suggested that the authors can cite some important charts in these literatures.
Response 2: Thanks for this suggestion. We have cited some important graphs and tables from the literature throughout the manuscript.
Comments 3: It is suggested that the authors can summarize the key genes and proteins related to the diagnosis of achalasia found by different omics in Table Form.
Response 3: As suggested by the reviewer, we have added the list of key genes and proteins in the tables.
Comments 4: How to integrate multi-omics to conduct research on achalasia for the main conclusions?
Response 4: We thank the reviewer for this comment. We have addressed this concept and included it in the revised manuscript in “red” in the SUMMARY AND FUTURE DIRECTIONS section.
Comments 5:
- As a clinical application, in fact, it is very concerned which key markers can be used for the diagnosis of achalasia?
Response 5a: Thank you for pointing this out. We have highlight this issue. It is in “red” in the SUMMARY AND FUTURE DIRECTIONS section.
- At the same time, there is a lot of concern about the feasibility of these markers as prognostic analysis, can the authors summarize and discuss the relevant aspects?
Response 5b: We have discussed this point in the manuscript and you can find it in “red” in the OMES, OMICS and MULTI-OMICS section.

Reviewer 2 Report
Comments and Suggestions for Authors
Review of the Paper: "Focus on Achalasia in the Omics Era"
Strengths:
- The manuscript offers a comprehensive review of achalasia through various omics approaches, providing an in-depth understanding of the disease.
- It effectively covers genomics, transcriptomics, metagenomics, viromics, and proteomics, detailing their contributions to understanding achalasia.
- The paper highlights the potential of integrating multi-omics data to uncover novel biomarkers and therapeutic targets.
- The discussion on the limitations of current omics technologies and future research directions is well-articulated.
Recommendations:
- Expand the discussion on the specific biological pathways and mechanisms implicated by the various omics findings. Include detailed descriptions of how genetic variants and molecular changes influence esophageal motility and contribute to achalasia
- Address potential sex-specific differences in achalasia pathogenesis and treatment responses. Discuss how sex-related factors might influence the omics data and disease outcomes
- Provide a clearer explanation of the statistical methods used in the referenced studies
- Explore the long-term stability and relevance of identified biomarkers and molecular changes
- Expand the discussion on translating omics findings into clinical practice.
- Consider adding a process diagram that encapsulates the paper's general methodology
- Provide more information regarding the datasets utilized in the studies discussed. Describe the sources, scope, and format of these datasets, as well as any preparation measures taken
- Highlight any research where experimental validation of omics findings has been achieved. Emphasize studies that have successfully translated omics data into clinical or experimental settings
- Discuss any ethical and legal issues associated with using omics data, particularly in relation to personalized medicine and patient privacy.
- Include a glossary of technical terms and acronyms used throughout the manuscript
Overall Assessment:
This manuscript provides a valuable and comprehensive review of achalasia through a multi-omics lens. The detailed explanations and integrative approach significantly enhance our understanding of this complex disease. Implementing the recommendations provided will further improve the manuscript's impact, clarity, and relevance, making it a notable contribution to the field of gastroenterology and multi-omics research.
Author Response
Reviewer 2
Strengths:
- The manuscript offers a comprehensive review of achalasia through various omics approaches, providing an in-depth understanding of the disease.
- It effectively covers genomics, transcriptomics, metagenomics, viromics, and proteomics, detailing their contributions to understanding achalasia.
- The paper highlights the potential of integrating multi-omics data to uncover novel biomarkers and therapeutic targets.
- The discussion on the limitations of current omics technologies and future research directions is well-articulated.
Comments 1: Expand the discussion on the specific biological pathways and mechanisms implicated by the various omics findings. Include detailed descriptions of how genetic variants and molecular changes influence esophageal motility and contribute to achalasia
Response 1: We thank the reviewer for appreciating our review. We caught the reviewer’s concern that has definitively increased the value of the manuscript. We have expanded the discussion of the requested topics that you can read in “red” in the SUMMARY AND FUTURE DIRECTIONS section belonging each specific omic. Moreover, as suggested, always in the same section you can find the description of how genetic variants influence esophageal motility and contribute to achalasia.
Comments 2: Address potential sex-specific differences in achalasia pathogenesis and treatment responses. Discuss how sex-related factors might influence the omics data and disease outcomes
Response 2a: Thanks for raising the point. We addressed about it in “red” in the SUMMARY AND FUTURE DIRECTIONS section and for Response 2b: in “red” in the OMES, OMICS and MULTI-OMICS section.
Comments 3: Provide a clearer explanation of the statistical methods used in the referenced studies
Response 3: We thank the reviewer for raising this issue, which we seek to clarify here. The articles included in this review analyzed their research results with different and appropriate tests using, when appropriate, tests ranging from "t-test" and "Fisher exact test", to "Cochran–Armitage" and "Cochran–Mantel–Haenszel". Generally, an uncorrected p-value <0.05 was considered nominally significant, while a p-value <1.4E-04 was considered significant after correction for multiple testing.
Comments 4: Explore the long-term stability and relevance of identified biomarkers and molecular changes
Response 4: We have, accordingly, emphasize this point in “red” in the OMES, OMICS and MULTI-OMICS section.
Comments 5: Expand the discussion on translating omics findings into clinical practice
Response 5: Thank you for pointing this out. As suggested, we broadened the discussion on the issue, increasing the significance of the findings. You can read it in “red” in the OMICS, OMICS and MULTI-OMICS section.
Comments 6: Consider adding a process diagram that encapsulates the paper's general methodology
Response 6: We have inserted the diagram as suggested.
Comments 7: Provide more information regarding the datasets utilized in the studies discussed. Describe the sources, scope, and format of these datasets, as well as any preparation measures taken
Response 7: We thank the reviewer for raising this issue and we try to clarify this point here. As highlighted in our review, current studies on achalasia are preliminary. Most reports are still based on simple case-control association studies that have yielded a limited number of genetic alterations and proteins, thus not querying public datasets, but rather analyzing their own data resulted from single-center cohorts or small consortia that analyze only collected samples or to which investigators have access.
Comments 8: Highlight any research where experimental validation of omics findings has been achieved. Emphasize studies that have successfully translated omics data into clinical or experimental settings
Response 8: We agree with this comment. Therefore, we have included this item in in “red” in the OMICS, OMICS and MULTI-OMICS section.
Comments 9: Discuss any ethical and legal issues associated with using omics data, particularly in relation to personalized medicine and patient privacy
Response 9: As highlighted we have discussed it in “red” in the OMICS, OMICS and MULTI-OMICS section.
Comments 10: Include a glossary of technical terms and acronyms used throughout the manuscript
Response 10: Done. Gene and protein names were not included in the glossary. You can find them throughout the manuscript and in the table captions.
Overall Assessment:
This manuscript provides a valuable and comprehensive review of achalasia through a multi-omics lens. The detailed explanations and integrative approach significantly enhance our understanding of this complex disease. Implementing the recommendations provided will further improve the manuscript's impact, clarity, and relevance, making it a notable contribution to the field of gastroenterology and multi-omics research.
Round 2
Reviewer 2 Report
Comments and Suggestions for Authors
Thank you for addressing the comments. The revisions have significantly enhanced the manuscript in expanding the discussion of biological pathways, sex-specific differences, statistical methods, and translational potential. The inclusion of the process diagram and glossary adds further clarity. I believe the manuscript is ready for publication.
Author Response
We have fixed the suggested points